# Novel Insights into the Role of HDL-Associated Sphingosine-1-Phosphate in Cardiometabolic Diseases

**DOI:** 10.3390/ijms20246273

**Published:** 2019-12-12

**Authors:** Elena M. G. Diarte-Añazco, Karen Alejandra Méndez-Lara, Antonio Pérez, Núria Alonso, Francisco Blanco-Vaca, Josep Julve

**Affiliations:** 1Institut de Recerca de l’Hospital de la Santa Creu i Sant Pau, and Institut d’Investigació Biomèdica Sant Pau (IIB Sant Pau), 08041 Barcelona, Spain; ediarte@santpau.cat; 2Departament de Bioquímica i Biologia Molecular, Universitat Autònoma de Barcelona, 08193 Bellaterra (Barcelona), Spain; aperez@santpau.cat; 3Centro de Investigación Biomédica en Red (CIBER) de Diabetes y Enfermedades Metabólicas Asociadas, CIBERDEM, 28029 Madrid, Spain; nalonso32416@yahoo.es; 4Servei d’Endocrinologia, Hospital de la Santa Creu i Sant Pau, IIB Sant Pau, 08041 Barcelona, Spain; 5Servei d’Endocrinologia, Hospital Universitari Germans Trias i Pujol, Badalona, 08916 Barcelona, Spain; 6Servei de Bioquímica, Hospital de la Santa Creu i Sant Pau, IIB Sant Pau, 08041 Barcelona, Spain

**Keywords:** sphingolipids, apolipoprotein M, diabetes mellitus, insulin signaling, cardiac function, cardiomyocyte homeostasis, heart failure

## Abstract

Sphingolipids are key signaling molecules involved in the regulation of cell physiology. These species are found in tissues and in circulation. Although they only constitute a small fraction in lipid composition of circulating lipoproteins, their concentration in plasma and distribution among plasma lipoproteins appears distorted under adverse cardiometabolic conditions such as diabetes mellitus. Sphingosine-1-phosphate (S1P), one of their main representatives, is involved in regulating cardiomyocyte homeostasis in different models of experimental cardiomyopathy. Cardiomyopathy is a common complication of diabetes mellitus and represents a main risk factor for heart failure. Notably, plasma concentration of S1P, particularly high-density lipoprotein (HDL)-bound S1P, may be decreased in patients with diabetes mellitus, and hence, inversely related to cardiac alterations. Despite this, little attention has been given to the circulating levels of either total S1P or HDL-bound S1P as potential biomarkers of diabetic cardiomyopathy. Thus, this review will focus on the potential role of HDL-bound S1P as a circulating biomarker in the diagnosis of main cardiometabolic complications frequently associated with systemic metabolic syndromes with impaired insulin signaling. Given the bioactive nature of these molecules, we also evaluated its potential of HDL-bound S1P-raising strategies for the treatment of cardiometabolic disease.

## 1. Introduction

Heart disease is one of the leading causes of morbidity and mortality in patients with diabetes. Heart failure (HF) is a complex clinical syndrome characterized by the manifestation of structural and functional cardiac alterations that impair heart function. Its diagnosis is particularly difficult in early stages or when its symptoms are not specific, as they can be confounded with other cardiac complications [1].

The prevalence of HF is elevated in patients with diabetes mellitus [2,3]. An increased risk of HF persists in patients with diabetes mellitus even after adjustment for well-established risk factors, including hypertension and coronary heart disease (reviewed in [4]). This particular form of diabetes-mellitus-related cardiomyopathy has been referred to as diabetic cardiomyopathy (DCM) [5,6].

Intracellular accumulation of ceramides may contribute to the pathogenesis of cardiac dysfunction [7]. Conversely, sphingosine-1-phosphate (S1P) has been revealed to display antagonistic actions toward ceramide and exert favorable actions on cardiomyocyte physiology and survival. The intracellular contents of these sphingolipids are finely regulated and re involved in cellular lifespan.

The plasma concentration of S1P is often altered in diabetic patients and is related to changes in their lipoprotein profiles [8,9]. A line of research has focused on high-density lipoprotein (HDL)-bound S1P [10]. An array of HDL cardiovascular actions has been demonstrated to be mediated by the lipoprotein-bound S1P interaction with cells, including those of endothelial cells, macrophages, and cardiomyocytes [10,11]. HDL are dysfunctional in diabetic patients [12], and conceivably, an altered HDL content in S1P could be in part responsible for diabetes mellitus-related cardiomyopathy.

S1P may have a role in many human diseases. Its evaluation as a circulating biomarker with a predictive value for HF in patients with diabetes mellitus remains incomplete. This review aimed to update the relationship of S1P with the occurrence of cardiometabolic diseases. The current knowledge on the potential impact of HDL-bound S1P-raising strategies for the treatment of cardiac outcomes is also revised.

## 2. DCM: A Risk Factor for HF

Clinical DCM may take several years to develop, and its progress appears to be directly related to diabetes duration [13]. However, it is not easily detectable at early stages, especially in asymptomatic diabetic patients.

DCM is currently defined as an asymptomatic subclinical left ventricular diastolic dysfunction. In fact, its diagnosis is currently based on the detection of cardiac hypertrophy and diastolic dysfunction by 2D echocardiography [14]. The diagnosis of DCM-related structural and/or functional cardiac alterations by imaging, especially at the initial stages of this syndrome, is difficult [15]. The latter is important since left ventricular diastolic dysfunction may fatally worsen systolic dysfunction [16]. When this occurs, cardiac dysfunction may inexorably result in irreversible HF [15].

Diastolic dysfunction may be defined by the detection of a preserved left ventricular ejection fraction with reduced diastolic filling, the prolongation of isovolumetric relaxation and increased atrial filling [17]. Nevertheless, recent evidence from independent clinical studies of extended diabetic cohorts has shed doubts about the usefulness of diastolic dysfunction as a first marker of subclinical stages of DCM [18].

The physiopathology of DCM is complex and multifactorial [18,19]. Cardiac lipotoxicity is regarded as one of its leading mechanisms of cardiomyocyte dysfunction [7]. Cardiac lipotoxicity may be triggered by a steady cellular oversupply of free fatty acids, mainly due to the enhanced mobilization of triglyceride stores from adipose tissue, which overrides cardiomyocyte metabolic and storage capacities and leads to impaired mitochondrial function and increased oxidative stress. Notably, under these circumstances, enhanced cardiac free fatty acid uptake not only leads to myocardial triglyceride accumulation (i.e., cardiac steatosis) but leads to their transformation into different bioactive species, including sphingolipids (i.e., ceramides, S1P), which can eventually affect cardiac cell lifespan.

## 3. S1P Determines Cardiomyocyte Fate

### 3.1. S1P Is an Intracellular Component of the “Sphingolipid Rheostat”

Sphingolipid metabolism (i.e., synthesis and catabolism) is tightly regulated. Sphingolipids are synthesized de novo in the endoplasmic reticulum (ER) (Figure 1). In particular, cellular content of S1P is mostly controlled by enzymes needed for synthesis and degradation, which are present in most mammalian cells, including cardiomyocytes [20]. Ceramides are the main synthetic precursors of S1P and other intracellular sphingolipids, including sphingosine and sphingomyelin [21]. In the ER, ceramides can be converted into sphingosine by neutral ceramidases, which can subsequently be phosphorylated by two different sphingosine kinases (SphKs; isoforms 1 and 2) to generate S1P [22] (Figure 1). Sphingolipids may also be transformed by different enzymes present in biological membranes. In this regard, two types of sphingomyelinases may hydrolyze sphingomyelin yielding ceramide [23]. S1P may be degraded by dephosphorylation by two S1P phosphatases (SPPs) to regenerate Sph, or cleaved by the S1P lyase, generating ethanolamine-phosphate and hexadecenal to leave the sphingolipid metabolic pathway [24,25] (Figure 1).

In a myocardial context, intracellular S1P coexists with other sphingolipid species, sphingosine and ceramide, and its relative content led to the proposal of the “sphingolipid rheostat” hypothesis (Figure 1), which suggests that the relative levels of S1P, and those of sphingosine and ceramide, may be important determinants of cell fate [26,27,28,29].

### 3.2. S1P Modulates Cellular Physiology

S1P is not simply an intermediate of sphingolipid metabolism. S1P is fairly ubiquitous and is also present at relatively elevated levels in plasma; however, only a small percentage (2%) has been regarded as active under physiological conditions [30].

S1P may act either directly within the same cell, where it is readily produced, or indirectly by being transported to the extracellular milieu to regulate the cellular physiology of the same cell or other cell types via autocrine/paracrine or ‘endocrine’ mechanisms [31], respectively, in a process defined as inside-out signaling by S1P [32] (Figure 1 and Figure 2).

The impact on cell fate of S1P-mediated activation is rather complex. S1P signaling was first described as an intracellular second messenger that may regulate cell growth, survival, and other functions by acting on as yet unknown intracellular effectors in a receptor-independent manner [33] (Figure 1). Nonetheless, whether and how such S1P mechanisms contribute to either vascular or cardiomyocyte physiology remains unknown. Additionally, S1P may also bind to G protein-coupled receptors (S1PRs), providing evidence for a role for this sphingolipid as a first messenger [34] (Figure 1 and Figure 2). The action of S1P is dependent on the tissue, mainly due to cell-specific determinants, including the overlapping distribution patterns of different S1PRs, intracellular downstream S1P signaling, or both [35] (Figure 2). The S1P–S1PR axis may also target many protein regulators and signaling molecules, including histone deacetylases, protein kinases, and transcription factors [36,37,38].

There are five subtypes of S1PRs. Isoforms 1–3 of S1PR are present in most organs/tissues, whereas the expression of S1PR4 predominates in hematopoietic and lymphatic tissues, and S1PR5 is mainly present in the central nervous system tissues. Notably, their tissue and cell-specific expression pattern sheds light on how this S1P may initiate an array of multiple signaling cascades, and hence differentially influences cell response. In turn, individual S1PRs may simultaneously transduce S1P signaling to one or more different G proteins downstream of each receptor; thus, accounting for a diverse variety of responses to S1P [39,40,41,42]. S1P-mediated transduction may depend on the type of second messenger, i.e., cAMP, inositol phosphates, or calcium, or downstream pathways engaged, including extracellular regulated kinases (ERKs) or Rho. In this regard, S1PR1 couples exclusively to G protein i (Gi), whereas S1PR2 and S1PR3 may couple to multiple G proteins, including Gi, G12/13 and Gq [43,44,45,46,47,48,49,50].

On a cardiac scope, isoforms 1–3 of S1PR are predominantly expressed in the adult rodent heart, with S1PR1 being the most abundant [51], whereas the relative abundances of S1PR1 and S1PR3 are similar in adult human hearts, with S1PR2 being expressed to a lesser extent. In particular, S1PR1 is abundantly expressed in ventricular, septal, and atrial cardiomyocytes and in endothelial cells of cardiac vessels [41]. However, which of these S1PRs, is the most relevant in cardiac protection has not yet been established, but experimental evidence suggests that a combined action of all of them may be crucial in cardiomyocyte physiology.

## 4. The Effect of S1P on the Cellular Mechanisms Involved in Cardiac Remodeling and Dysfunction

S1P signaling has emerged as an important regulator of cardiac and vascular homeostasis [10] and has reportedly been related to the pathogenesis of multiple cardiovascular outcomes, including coronary artery disease, atherosclerosis, myocardial infarction, and/or HF [11,52,53,54,55,56,57,58,59,60,61,62,63]. In addition to the effects of S1P on atherosclerosis, which is regarded as a major mechanism involved in ischemic cardiovascular burden, a protective role of S1P on injured myocardium elicited by nonischemic causes has also been suggested [64,65,66].

S1P–S1PR signaling targets a broad array of intracellular metabolic pathways controlling many cellular processes involved in the homeostasis of multiple cell types (Figure 3). In a cardiac cell context, differential characteristics of S1PRs may potentially contribute to intrinsic myocardial functions of S1P, and the relative abundance of different S1PRs may be influenced in several cardiac pathologies. Indeed, cardiac S1PR1 appears to be downregulated in experimental models of HF, whereas its overexpression alleviates HF after myocardial infarction [67]. S1PR2 and S1PR3 also play a significant role in protecting cardiomyocytes from ischemic-reperfusion (I/R) damage in vivo [68]. Overall, these observations may reveal S1P/S1PR-specific cardioprotective mechanisms in regulating cardiac physiology, and hence offer novel therapeutic strategies by targeting this axis (Section 7). However, the specific S1PRs involved in cardioprotective effects have not yet been dissected, with compelling evidence potentially suggesting that S1PR1, S1PR2, and S1PR3 might work together [69].

Cardiac alterations could also be induced by metabolic stress, such as that present during DCM development [70]. To our knowledge, no study has directly explored the impact of S1P on nonischemic causes of cardiomyopathy. As previously mentioned, in most studies, the cardioprotective impact of S1P on either cardiomyocytes or the myocardium has been assessed under oxidative stress conditions derived from experimental acute/chronic ischemia or I/R injuries, whereby S1P elevations produce pro-survival effects on cardiomyocytes under hypoxic conditions in both in vitro and in vivo models [27,71].

The main cellular mechanisms that are commonly involved in cardiac remodeling and function during DCM progression are depicted in Figure 3 [19]. These are thought to mainly be triggered by the metabolic consequences of dysfunctional insulin signaling [72], which leads to a series of events that results in cardiac lipotoxicity (Section 2). Lipotoxicity can aggravate chronic cardiac and vascular dysfunction (Figure 3) [7,19,70].

### 4.1. Oxidative Stress

Reactive oxygen species (ROS) are highly reactive molecules that are produced in the mitochondria as byproducts of oxidative phosphorylation [73]. When in excess, their accumulation overwhelms cellular antioxidant and repair mechanisms, leading to pathogenic cellular damage [74]. The accumulation of misfolded proteins in the ER is enhanced under oxidative stress, contributes to ER stress, and compromises cellular homeostasis [75].

Oxidative stress contributes to cardiac remodeling and dysfunction [7]. Current evidence supports the concept that altered synthesis of molecular determinants of “Sphingolipid rheostat” determine the cellular response to oxidative stress in cardiomyocytes in vitro [76] and in vivo [77]. Consistent with this view, increased levels of S1P are associated with decreased ROS production [78].

Oxidative stress also influences enzymes involved in S1P synthesis, i.e., Sphk1/2, and hence alters the “sphingolipid rheostat.” For instance, the inhibition of SphK1 is enhanced in conditions of elevated cellular oxidative stress in cardiomyocytes [77], potentially due to its increased misfolding. The ROS-mediated inhibition of SphK1 might be considered a cellular mechanism to partly explain ROS-induced apoptosis. In support of this, in previous studies, a decrease in SphK1 has been regarded as a key event in the generation of apoptotic ceramide [79,80], possibly by altering the cellular “sphingolipid rheostat.”

### 4.2. Inflammation

S1P favorably regulates inflammatory processes in a variety of organs. Conceivably, S1P could also exert similar beneficial anti-inflammatory actions in other phenotypes, such as that found by the failing heart [81]. In this regard, S1P has atheroprotective actions during the progression of atherosclerosis via a multitude of mechanisms involving S1PR1–4 [82].

Treatment with S1P prevents the inflammatory activation of monocytes in vitro [83]. Nevertheless, its potential effect in vivo using animal models has led to controversial results. For instance, elevations in S1P produce favorable reductions in infarct size in a mouse model of I/R injury [84], and this effect was strongly dependent on S1PR3 signaling as it was completely abolished in S1PR3-deficient mice. In contrast, S1PR3 deficiency does not produce any effect on the atherosclerotic lesions of ApoE-deficient mice [85]. Importantly, the number of macrophages in the aortic lesions and other beds (peritoneal) of double S1PR3- and ApoE-deficient mice is lower than that of ApoE-deficient mice. Notably, in the same study, the absence of S1PR3 signaling also promoted smooth muscle cell proliferation and neointima formation in atherosclerotic lesions after carotid ligation. Overall, these findings suggest that the anti-atherogenic action of S1P might be partly dependent on S1PR3-specific actions in different cell types.

In contrast to S1PR3, the ablation of S1PR2 signaling favorably reduced the extent of atherosclerotic lesions, macrophage recruitment, and systemic anti-inflammatory response [86], thereby suggesting that S1PR2 action might be pro-atherogenic. However, in independent studies, the induction of S1PR2 expression abrogates vascular smooth muscle cell proliferation [87,88], which is enhanced during atherogenesis.

### 4.3. Mitochondrial Dysfunction

Mitochondrial dysfunction is a key feature of DCM and is present in cardiac tissue from diabetic patients and animal models of diabetes. Mitochondrial dysfunction has been related to elevated intracellular levels of ROS [89] which in turn may contribute to cardiomyocyte damage [90]. In this context, S1P has been reported to favorably modulate mitochondrial function in damaged cardiomyocytes [91,92], at least in part by promoting cytochrome-c oxidase assembly [93].

### 4.4. Calcium Handling

Calcium metabolism is crucial for myocardial contractility and modulates heart pump function [94]. S1P regulates calcium metabolism in cardiomyocytes [39,95] and the ionic status in cells of the sinoatrial node, which reduces heart rate [39,96], and has conceivably been reported to promote ventricular contraction and blood pressure [39].

Alterations in Ca^2+^ handling, in particular, Ca^2+^ release from the sarcoplasmic reticulum, which is regarded as the main reservoir of intracellular Ca^2+^ in cardiomyocytes, may result in cytosolic Ca^2+^ overload and cardiac dysfunction as observed in I/R [97] and HF [98]. Recent data show that the continuous induction of S1P signaling may confer cardiac protection against I/R injury [91,99] and ischemic preconditioning [100] by contributing to intracellular Ca^2+^ homeostasis.

### 4.5. Cardiomyocyte Hypertrophy

The role of S1P signaling in cardiomyocyte hypertrophy, and thus, cardiac remodeling, is controversial. Indeed, S1P has been reported to have no effect on hypertrophy in vitro [101], promote the hypertrophy of rat and mouse cardiomyocytes via S1PR1 [40], or attenuate cardiomyocyte hypertrophy [38,102] in independent in vitro studies. Such controversy is also present when evaluating the effect of stimulating the S1P axis in vivo. In this regard, S1P has been reported to ameliorate cardiac hypertrophy by either inhibiting the action of histone deacetylase-2, which is a pro-hypertrophic factor in the heart [38], or reducing the expression of other hypertrophic genes, including *atrial natriuretic factor*, skeletal muscle and cardiac actin, myosin heavy chain, and brain natriuretic peptide [102]. Despite this, the activation of S1P/S1PR1/SphK signaling has been previously shown to promote cardiomyocyte hypertrophy in response to several adverse stimuli [40]. Consistently, cardiac S1P/S1PRs/SphK signaling is induced in postmyocardial infarction hearts and contributes to chronic cardiac remodeling in vivo [103]. Such controversy could, at least in part, be explained by the induction of S1PR1 signaling in cardiac fibroblasts [104], which might eventually be involved in worsening both cardiac hypertrophy and fibrosis.

### 4.6. Autophagy

A physiological level of autophagy is essential for removal of dysfunctional mitochondria and maintaining cellular homeostasis. Several reports suggest a role for autophagy in cardioprotection [105]. Although autophagy can be protective during cardiac injury [106] and myocardial damage [107], altered activation can also lead to cardiac dysfunction [106,108,109] and HF [110].

An active role for S1P has been suggested to modulate autophagy and cell survival [111,112]. Indeed, S1P has been reported to protect cardiac function by inhibiting autophagy during myocardial damage [102]. Supporting this concept, the induction of SphK1, which increases intracellular levels of S1P, induces autophagy-related survival [112]. Conversely, autophagy-related death is mainly controlled by ceramide.

DCM is associated with the suppression of cardiac autophagy [113,114,115]. Whether the induction of autophagy prevents the development of DCM has been poorly explored; however, recent data support that the induction of autophagy may be protective [116].

### 4.7. Apoptosis

Intracellular elevations in S1P produce pro-survival anti-apoptotic effects in vitro and in vivo [20,27,117]. Indeed, the induction of Sphk1 has pro-survival properties, whereas either the overexpression or downregulation of SphK2 induces apoptosis [31]. Conceivably, increased levels of S1P generated by SphK1 activation can be exported outside the cells, where S1P can bind to its receptors to exert favorable pro-survival effects in an autocrine and/or paracrine manner [31] in cardiomyocytes and other cell types [118,119].

The selective inactivation of SphK1 and 2, which results in reductions in intracellular levels of S1P, in adult cardiomyocytes accelerates cell death by hypoxia-induced apoptosis [118]. Consistently, Sphk1 inhibition, promoted by elevated oxidative stress, mediates cardiomyocyte apoptosis [77]. Additionally, the inhibition of an endogenous Sphk1 inhibitor, known as four-and-a-half LIM domain 2, conferred protection from apoptosis to cardiomyocytes [120]. S1P can also act intracellularly to suppress apoptosis in an S1PR-independent way [31]. However, the identification of intracellular S1P targets remains elusive.

### 4.8. Fibrosis

Myocardial fibrosis is one of the main structural changes in HF development [121,122,123]. The role of S1P in mediating myocardial cell fibrosis is controversial [124], partly due to limitations in the experimental models used.

Cardiac SphK1/S1P/S1PR1 signaling is a major mediator of transforming growth factor-β (TGF-β)-stimulated myofibroblast activation and collagen deposition, which eventually contributes to cardiac fibrosis and remodeling [125]. SphK1, a main determinant of S1P intracellular levels, has been shown to prevent the occurrence of cardiac fibrosis under normal physiological conditions [124]. Nevertheless, the overexpression of this enzyme enhances cardiac fibrosis in vivo [104,126]; hence, contributing to observed myocardium degeneration. Similar to cardiomyocytes, cardiac fibroblasts can change from a static to a proliferative and migratory state under adverse conditions, which may eventually lead to myocardial fibrosis [127], and conceivably, worse HF. Overall, these findings suggest the participation of cell-specific mechanisms involved in the activation of S1PR signaling, and therefore acting against myocardium fibrosis.

In addition to S1PR1, the role of other S1PR signaling pathways might also influence myocardium fibrosis. In this regard, S1PR3-deficient heart cells abrogate myocardium fibrosis in SphK1-overexpressing mice [126], suggesting that S1PR3 might be a potential target to treat pathological fibrosis development.

Altogether, S1P, S1PRs, and SphK are interrelated and may play a role in the development and pathogenesis of cardiac fibrosis. However, the molecular underlying mechanisms are not completely understood. Whether S1P-based therapies may favorably influence myocardium fibrosis remains to be defined.

### 4.9. Endothelial Dysfunction

Endothelial cells are covered by surface proteoglycans that form part of the barrier functions of the blood-tissue interface [128]. Endothelial cells are a main cell type in the vasculature and contribute to the maintenance of vascular tone and cardiac repair [129]. Vascular leaks of fluid from the plasma to the interstitial space contribute to inflammation and myocardial dysfunction.

S1P induces multiple effects on the vasculature [130], in part by the variable abundance of specific S1P receptors in different cell types (such as endothelial cells, macrophages, and smooth muscle cells).

S1P and its main receptors in endothelial cells (predominantly S1PR1 and S1PR3) play a central role in S1P signaling in the maintenance of endothelial barrier integrity and preventing micro- and macrovascular damage [130,131,132]. Indeed, the transactivation of S1PR1/3 by S1P induces endothelial nitric oxide synthase (eNOS), thereby promoting nitric oxide production and vasodilation [133,134,135]. Supporting this, the incubation of cultured endothelial cells with S1P in conditions mimicking impaired vessel perfusion has been found to prevent proteoglycan loss in vitro [136], whereas S1PR1 antagonism abrogates the protection of glycocalyx degradation by S1P [136].

## 5. S1P in Circulation: HDL, More than a Cargo of S1P

Lipidomic analysis revealed the presence of significant circulating amounts of over 200 species of sphingolipids [137,138,139]. A significant proportion of plasma sphingolipids are associated with plasma lipoproteins. Moreover, it has been suggested that sphingolipid transport on either ApoB-containing lipoproteins or HDL is also required for delivery to other different tissues. Importantly, sphingolipids may also be transported by other plasma vehicles such as albumin, but this nonlipoprotein carrier component may be less potent than HDL-bound S1P to trigger long-term responses in cells [140,141]. Consistent with this finding, the lifespan of S1P in HDL is longer (four times) than that in albumin [141]. These findings further suggest that HDL is a stable reservoir of active S1P in the circulation [11]. Albumin has been proposed as another reservoir for S1P, potentially preventing the overstimulation of S1PRs [11].

The presence of sphingolipids in plasma lipoproteins also suggests that they may function as carriers enabling the distribution of S1P to more distant tissues. The transport into lipoproteins may possibly involve different mechanisms [142]. First, these compounds can associate with early lipoprotein constituents (ApoB or ApoA-I) during intracellular lipoprotein biogenesis and assembly before being released into the circulation. Additionally, their exchange during lipoprotein remodeling (partly explained by the action of plasma lipid transport proteins, such as phospholipid transfer protein or cholesterol ester transfer protein, see below) may allow S1P to redistribute among lipoproteins [143,144]. Conceivably, S1P may be degraded to other sphingolipids or even synthesized from sphingolipids present on lipoproteins and be transported to target tissues different from their site of synthesis.

### 5.1. HDL-Bound S1P and Apolipoprotein (Apo)M

S1P is present in circulation at high nanomolar concentrations. The plasma levels of S1P in healthy subjects mostly range between 200 and 1000 nM [30], and it is relatively more abundant in plasma than in tissues [145,146,147,148]. The amphipathic nature of this molecule makes it impossible to find it unbound in plasma.

In healthy humans, S1P is mainly present in HDL (>55%) [30,149,150] and to a lesser extent in other lipoproteins, mainly LDL (~10%) and VLDL (2–3%), with a remaining portion bound to albumin (35%) [139,151,152]. Notably, since S1P binds to all five S1PR molecules to activate signaling, with relatively low nM K_d_ values [30], it has been proposed that a significant proportion of S1P in HDL may be considered biologically inactive, and hence, not accessible for signaling purposes [10].

Several proteins are involved in the modulation of plasma levels of S1P. S1P may be effluxed to extracellular plasma acceptors, i.e., either plasma nonlipoprotein or lipoprotein carriers [30]. In this regard, S1P can be effluxed to plasma HDL [153], in a process that is mainly mediated by the action of apolipoprotein (Apo)M [154].

ApoM, a member of the lipocalin protein superfamily, has been proven to be critical for S1P association with HDL in vitro [155] and in vivo [156]. Plasma levels of ApoM are approximately ~0.9 µM. It has been estimated that approximately 95% is bound to HDL [157], being relatively more elevated in the small, dense, protein-rich HDL-3 subclass [150,158,159,160,161,162]. ApoM is mainly synthesized by the liver and is involved in the transport of S1P in plasma [163,164], where it is involved in S1P protection against extracellular degradation. Supporting this view, experimental ApoM deficiency in vitro and in vivo leads to concomitant decreases in plasma S1P [156]. However, some data do not favor ApoM as the main vehicle of S1P [165,166]. Indeed, the ratio of ApoM-to-HDL particles is much lower than that of S1P-to-HDL particles [11,167]. Other molecules, including oxidized phospholipids and retinol, may compete with S1P for the same lipophilic pocket of ApoM and may contribute to reducing the amount of S1P transported [167] (Figure 4). Plasma levels of S1P and ApoM do not correlate in patients with genetic HDL deficiencies [165]. Concomitantly, circulating levels of ApoM and the incidence of cardiovascular disease are not correlated [168]. Additionally, and as revealed in ApoM-deficient mice, the concentration in plasma of S1P is only approximately 50% lower than that of wild-type mice [156], suggesting the contributions of other plasma constituents in maintaining the plasma levels of this metabolite. In a recent report, liver-specific ApoM overexpression in mice produced high S1P-enriched HDL in plasma, despite showing similar total HDL cholesterol levels to wild-type mice [164]. In these ApoM transgenic mice, S1P synthesis was enhanced, thereby suggesting that the export of S1P from hepatocytes may be ApoM-dependent.

### 5.2. Transporters Involved in Extracellular S1P Efflux and Signaling

#### 5.2.1. Spinster2

Spinster2 (Spns2) is a cell membrane protein that belongs to the major facilitator superfamily [169]. It allows S1P transport out of cells and its release to the plasma [170]. Consistently, Spns2 deficiency in vivo leads to concomitant decreases in S1P in plasma [171,172]. Conceivably, Spns2 may also influence plasma levels of HDL-bound S1P (Figure 2). Notwithstanding, evidence supporting the exact contribution of Spns2 in the export of S1P from cells has not yet been reported.

#### 5.2.2. ATP-Binding Cassette (ABC) Transporters

HDLs may mediate the efflux of S1P from circulating erythrocytes by virtue of contacting the latter with their plasma membrane [56,173] (Figure 2). Whether this process is mediated by passive diffusion mechanisms or involves active transportation is still unknown. However, S1P transport from cells has also been proposed to be mediated by specific ABC transporters [170]. Consistently, S1P release may be inhibited by a nonspecific inhibitor of ABC transporters [174,175]. Other studies report the contribution of ABCC1 and ABCG2 to the S1P cargo carried by HDL from different cell types [11,176,177,178,179,180]. Despite this, no evidence has been reported showing a direct interaction between HDL and these ABC transporters. Other ABC transporters (i.e., ABCA1 and ABCG1), which stimulate the cholesterol efflux [181], may also promote S1P enrichment of HDL-like lipoproteins [182,183]. More recently, it has been reported that ApoA-I promotes S1P release from endothelial cells via ABCA1 [184]. Despite this, the role of this transporter in plasma levels of S1P remains controversial. For instance, while patients with mutations in ABCA1 present lower plasma levels of S1P [165], the deficiency of this transporter in in homozygous Tangier patients or mice does not cause significant reductions in S1P [185,186]. Therefore, the mechanism underlying S1P release to plasma lipoproteins still needs to be elucidated.

#### 5.2.3. Scavenger Receptor Class B Type I (SR-BI)

The interaction of certain components of HDL on the cell membranes is facilitated by SR-BI [187]. Recent evidence supports the concept that HDL-bound S1P signaling may require both S1PR and SR-B1 receptors [11]. Whether HDL-bound S1P signaling is mediated by facilitating an effective interaction between HDL-bound S1P and S1PRs or coordinating receptor-mediated signaling is still a matter of debate [11] (Figure 2).

The abrogation of SR-BI action has been suggested to attenuate HDL-bound S1P associated biological effects and signaling pathways [134,188]. In support of this, it has been reported that HDL-bound S1P stimulates the molecular interaction of both S1PR1 and SR-BI receptors to boost S1P-mediated changes at sites where HDL is docked by this receptor [189]. Data from that study strongly suggest that the presence of SR-BI, and perhaps other HDL receptors/HDL binding proteins, might selectively and specifically direct HDL-bound S1P-mediated signaling via S1PRs in a way that differs from that occurring when S1PRs are activated by albumin-bound S1P. SR-BI has recently been reported to be involved in S1P/S1PR2-mediated inflammation in endothelial cells [190], suggesting that SR-BI may not just dock HDL to cell membrane but might contribute to the establishment of a molecular crosstalk between ApoA-I/SR-BI and S1P/S1PR signaling to induce diverse cellular responses (Figure 2).

### 5.3. Proteins Involved in Plasma HDL Remodeling

As components of HDL, plasma levels of S1P and ApoM might also be influenced by two plasma proteins during the remodeling of these lipoproteins; i.e., phospholipid transfer protein (PLTP) or cholesterol ester transfer protein (CETP). PLTP may transfer S1P from isolated erythrocytes to HDL [56]. Consistently, a deficiency in PLTP significantly reduces S1P in vivo [143]. On the other hand, CETP may also modulate the distribution of S1P among lipoproteins, affecting its biological functions. The overexpression of CETP in mice does not influence the levels of either total plasma S1P or its main carrier; i.e., ApoM [144]. Intriguingly, S1P distributions can be shifted from HDL to ApoB-containing lipoproteins [144]. Although the mechanisms of S1P redistribution in the latter study were not deeply evaluated, the authors suggested that the CETP-mediated modulation of lipid profiles might be a main determinant of lipoprotein S1P content. It could be hypothesized that the failure of CETP inhibitor-based therapies to improve cardiovascular risk might be at least in part attributable to alterations in the S1P cargo of ApoB-containing lipoproteins.

## 6. S1P in Cardiovascular Disease: A Role for HDL?

Current evidence suggests that S1P might, at least in part, be regarded as the mediator of many of the favorable cardiovascular effects of HDL, involving cellular actions such as the suppression of oxidative/inflammatory processes, apoptosis, and endothelial dysfunction [191,192]. Supporting this notion, HDL-bound S1P may be determinant of S1P biological action by facilitating the presentation to specific receptors [189] (Figure 2). In this regard, ApoM, by virtue of its ability to bind S1P, may also contribute to the beneficial cardioprotective effects of HDL [10]. Indeed, ApoM overexpression protects against atherosclerotic development [193]. Overall these data suggest that S1P-bound ApoM, rather than total S1P bound to other carriers, such as albumin, could be critical in conferring protection against atherosclerosis [11]. Although the content of S1P in the HDL from mice overexpressing ApoM was not determined [193], in line with recent data, it might be expected that the circulating levels of HDL-bound S1P would be concomitantly increased [164]. Consistent with this concept, experimental strategies designed to neutralize HDL-bound S1P, render HDL less functional [11,30,56,141,194]. In a diabetic context, HDL with an elevated content of S1P conferred protection on endothelial cells via S1PR1/3 [195].

Several lines of evidence also support a role for HDL-bound S1P and/or ApoM in preventing HF. Indeed, the S1P content of HDL also prevents ventricular cardiomyocytes from developing induced apoptosis [196], strongly suggesting a role for circulating HDL-bound S1P in limiting cardiomyocyte death. Both HDL-bound ApoM and S1P attenuate endothelial cell death by blocking apoptosis in a cooperative manner acting via endothelial S1PR1 and S1PR3 [197]. Recent data suggest that endothelium-protective S1P has been related to the HDL-bound ApoM [156]. Nevertheless, in line with previous findings (Section 5.1), the favorable contribution of ApoM-independent HDL-bound S1P should not be ruled out.

Some evidence also supports the potential role for other S1P-independent mechanisms in explaining the anti-atherogenic protection by HDL-bound ApoM [193,198,199]. Indeed, ApoM, independently of its role as a carrier of S1P, promotes the formation of nascent HDL (i.e., preβ-HDL) [193,200] and stimulates cholesterol efflux [193,201], which is regarded as the first step of reverse cholesterol transport in vivo [193]. Additionally, ApoM may also protect LDL from oxidation [158,202].

## 7. Assessment of Circulating S1P in Cardiometabolic Diseases

Although diastolic dysfunction is regarded as a functional alteration in patients with DCM [13,14], the diagnosis of DCM by imaging is not easy, especially at the initial stages of this syndrome [15] (Section 2). In addition, there is a lack of specific biomarkers with which to diagnose diabetes mellitus-related cardiomyopathy. Therefore, the investigation of new noninvasive strategies for the diagnosis and prognosis of this condition represents a challenge in clinical practice.

Plasma or serum samples are currently used to determine S1P in different studies [10]. The presence of S1P in circulation and its distribution among blood components, preferentially HDL, along with the reported differences in plasma levels and redistribution among HDL subclasses in patients with metabolic syndromes, suggest this compound as a candidate biomarker (reviewed in [10]). However, some technical flaws obscure this potential as a biomarker. For instance, lipoprotein analysis requires the use of ultracentrifugation devices, which are not available in most clinical laboratories. There are different analytical methodologies available to determine S1P levels in plasma, including HPLC, LC-MS, and commercial ELISA kits, thereby yielding important differences among studies [10]. A comprehensive comparison among different methods is still pending and will be needed before application in a clinical setting. Additionally, plasma S1P usually appears normalized to different HDL components (e.g., ApoA-I or ApoM) or HDL mass in different studies [195,203,204], and this may also make data comparison difficult.

### 7.1. Circulating S1P/ApoM in Cardiac Diseases

The potential of S1P as novel, noninvasive indicator of coronary artery disease has been assessed in different reports. For instance, some studies have described that total S1P plasma levels may be reduced in patients with either myocardial infarction or coronary artery disease [52,205]. Circulating S1P has also been shown to be a strong predictor of occurrence and severity of coronary stenosis [206]. An inverse association between coronary artery disease and HDL-bound S1P has also been reported [52,192]. Interestingly, plasma levels of HDL-unbound S1P prevailed over HDL-bound S1P in patients with myocardial infarction and stable angina in one of these studies [52]. This suggests that S1P bound to HDL would confer anti-atherogenic protection. Consistently, decreased HDL-bound S1P has been related to coronary artery disease [207,208]. It is worth noting that in one of these studies the content of S1P in HDL was inversely related to the incidence of coronary artery disease independent of the plasma concentration of HDL cholesterol [192]. Likewise, decreased concentrations of both S1P and HDL-bound S1P have been reported in patients with coronary in-stent restenosis compared with healthy subjects [209]. Accordingly, HDL-bound S1P has been further revealed as an independent predictor of this coronary event in affected patients [209], but it fails to predict the severity of coronary heart disease burden [208].

ApoM prevents atherosclerosis in murine models [193,198,210]. Some evidence suggests that such ApoM-mediated atheroprotection could be partly due to the ability of this apolipoprotein to generate lipid-poor preβ-HDL, which is an acceptor of cellular cholesterol involved in the reverse cholesterol transport pathway, and it confers anti-oxidant protection [158,193,202,211] (Section 6). Unfortunately, the plasma concentration of total or HDL-bound ApoM was not examined in any of the abovementioned studies [52,192,205,206]. Despite this, ApoM plasma concentration did not differ in two separate case-control studies (i.e., FINRISKʼ92 and CCHS) [168]. On the other hand, even though plasma ApoM was reduced in patients with metabolic syndrome, it failed to predict subclinical atherosclerosis [210].

In summary, all these data suggest that S1P, rather than ApoM, could be a potential candidate biomarker for predicting subclinical cardiometabolic events.

### 7.2. Circulating S1P/ApoM in Patients with Diabetes Mellitus and Relationship with Cardiac Disease

The plasma concentration of S1P has been found to be either elevated [212,213] or reduced [214,215] in diabetic patients in independent studies. Plasma elevations of S1P have also been found in obese patients [216]. Interestingly, such elevations are directly related to adiposity in patients with dysfunctional glucose metabolism [216,217].

The S1P content of isolated HDL in diabetic patients has been analyzed in only a few reports. Plasma levels of HDL-bound S1P have been observed to be elevated in diabetic patients [195]. Notably, in [195] the concentration of HDL-bound S1P was normalized by the HDL mass. Therefore, the calculated levels of HDL-bound S1P could have likely been overestimated. HDL-bound S1P has been found to be inversely related to glycated hemoglobin (HbA1c) in patients with type 2 diabetes mellitus in an independent study [203], suggesting that glycemic control can affect the results obtained.

The plasma concentration of ApoM was decreased in a study with patients with type 2 diabetes mellitus [218] but was unchanged in another study [215]. Circulating ApoM did not differ in type 1 diabetes mellitus patients compared with nondiabetic subjects, but ApoM/S1P complexes were shifted from dense to light HDL particles in diabetic patients [204]. Nonetheless, the relationship of ApoM, if any, with cardiac dysfunction, was not eventually assessed.

Whether plasma levels of S1P may be useful in predicting increased risk for cardiovascular disease in diabetic patients is still unknown. However, compelling clinical evidence suggests that S1P may play a role in the modulation of key cellular processes involved in atherosclerosis burden. Indeed, S1P has been found to be directly related to the HDL-mediated ability to activate endothelial eNOS [214]; thus, reductions in plasma S1P might possibly contribute to endothelial dysfunction in diabetic patients. Of note, HDL isolated from patients with type 2 diabetes mellitus is less efficient in maintaining proper endothelial function, and notably, endothelial function is independent of plasma HDL cholesterol. Accordingly, plasma S1P has also been identified as an independent predictor against the development of coronary artery disease in patients with type diabetes mellitus [215].

## 8. Assessment of S1P–S1PR-Based Experimental Pharmacological Strategies in Ischemic Heart Disease

As the concept is that the induction of S1P signaling may influence heart function, its modulation has been considered a promising strategy to improve cardiac diseases [31,219].

### 8.1. Strategies Targeting S1P Signaling to Improve Cardiac Dysfunction: A Role for HDL?

A growing body of evidence suggests that dysfunctional HDL may be instrumental in reducing cardioprotection and suggests the involvement of HDL-bound S1P as a potential driving mechanism [10]. Indeed, the abrogation of HDL-bound S1P-mediated action by a specific neutralizing antibody (i.e., Sphingomab) [56], S1P removal by the delipidation of HDL [30], or reconstituted HDL without S1P reloading [194] induced loss of HDL function. These findings suggest that S1P-raising strategies could be considered therapeutic strategies.

Circulating HDLs are dysfunctional in diabetic patients [220]. Consistent with this view, S1P content in the HDL from diabetic patients has been reported to be inversely and directly correlated with glycation and cardiomyocyte survival, respectively [203]. Interestingly, the therapeutic replenishment of HDL with S1P has been reported to completely prevent and even restore the anti-inflammatory capacity of HDL isolated from patients with coronary artery disease [219]. Thus, the addition of S1P to dysfunctional diabetic HDL might conceivably restore its cardioprotective potential (Figure 4).

S1P-enriched, reconstituted (r)HDL induce enhanced vasorelaxation compared to control rHDL in aortas [162] and protect cardiomyocytes against apoptosis [196]. Similarly, HDL-bound S1P reduces cardiomyocyte apoptosis in postischemic myocardium, which in turn depends on NOS activity [84]. Supporting this, the activation of eNOS promoted by HDL from patients with metabolic syndrome, which are depleted of S1P, is decreased with respect to controls [221]. Moreover, protective function in endothelial cells is restored by S1P-replenishment of HDL from type 2 diabetes mellitus patients [222].

Several findings support a crucial role for HDL-bound ApoM in driving S1P bioavailability and endothelium protection. This is partly because HDL-associated S1P may be specifically bound to ApoM [156]. Not surprisingly, liver-specific ApoM overexpression raises the plasma concentration of HDL-ApoM/S1P [164] by eliciting S1P biosynthesis and secretion. In addition, ApoM also protects S1P from degradation, and hence, increases plasma S1P lifespan and activity [223]. It is also thought that ApoM, by delivering S1P to the S1PR1 receptor on endothelial cells, contributes to vascular protection [156]. In support of this, HDL-bound ApoM/S1P has been found to protect the endothelium from inflammation [224] and apoptosis [150]. These data may support the concept that the elevation of ApoM/S1P-enriched plasma HDL could also be a potential S1P delivery strategy to S1PR1 of extrahepatic tissues for cardioprotective purposes.

### 8.2. Pharmacological Approaches Targeting S1P Signaling

Studies in anemic patients receiving erythrocyte transfusion showed elevation in HDL-bound S1P [225]. Based on this, another approach involving S1P-preloaded erythrocyte transfusion has been proposed to elevate S1P in dysfunctional HDL in vivo [56]. This strategy is based on the ability of HDL to extract S1P from cell surfaces, including erythrocyte membranes (Figure 5). Although the mechanisms whereby S1P accumulates in HDL are under investigation, they may involve several putative transporters and plasma proteins involved in HDL remodeling. Consistent with previous data [173], S1P-bound erythrocyte transfusion in vivo predominantly and efficiently provides the plasma HDL fraction with S1P. Therefore, two different erythrocyte S1P-loading strategies were tested [56]. Indeed, autologous erythrocytes can be directly loaded with S1P or incubated with a specific inhibitor of S1P lyase (i.e., 4-deoxypyridoxine) to raise intracellular content of S1P ex vivo before being reinjected into mice (Figure 5).

Whether S1P was retained in the HDL after erythrocyte transfusion and led to improved cardioprotection was not eventually assessed. However, this strategy would open the door to novel therapeutic strategies using the autologous transfusion of S1P-preloaded erythrocytes to correct dysfunctional HDL (Figure 5). Thus, further research is warranted in this field.

There are at least two other potential strategies to modulate S1P signaling. One of them is based on the direct modulation of S1P synthesis by using specific activators/inhibitors of SphK [20]. Additionally, S1P signaling can be effectively modulated by influencing S1PR-mediated transduction to downstream effectors [226].

Drug development has mostly been directed to targeting S1P signaling [226]. This is because S1P actions are mainly mediated by S1PRs, which determine drug selectivity. Of these, S1PR1 was the first to be validated as a target. Compared with S1PR1, much less is known about the physiopathological role of other S1PRs [226].

The role played by S1P in the development of cardiovascular diseases makes S1PRs the most interesting pharmacological targets. In this regard, the mechanism of action and the efficacy and safety of several main S1PR1 modulators, i.e., FTY720 or SEW2871, have been evaluated in recent studies [31,226,227,228]. Several other compounds include drugs currently used in clinical practice or that are at different stages of clinical development.

#### 8.2.1. FTY720

The interest in S1P signaling effects increased with the discovery that FTY720 (fingolimod; 2-amino-2-(2-[4-octylphenyl]ethyl)-1,3-propanediol) modulates the immune system by modulating S1PR1. FTY720 is an oral medication approved by United States Food and Drug Administration in 2010 for the treatment of relapsing-remitting multiple sclerosis [31,229].

FTY720 is an immunosuppressant administered as a pro-drug, which, when phosphorylated by SphK2 in vivo [230], acts as a structural analog of S1P (Figure 6). Phosphorylated FTY720 (FTY720-monophosphate; FTY720-P) undergoes the action of the same phosphatases (SPP1 and SPP2) that dephosphorylate S1P and is transported out of cells by Spns2 [231,232]. FTY720 binds to all S1PRs, except S1PR2 [231,232,233,234]. Notably, and similarly to S1P, FTY720-P can be transported by the human Spns2 [232]. It also acts as a potent S1PR agonist, preferentially to S1PR1, and to a lesser extent, S1PR3 [235], thereby mimicking S1P responses.

In relation to the cardiovascular system, FTY720 blocks chronic inflammation, protects both the macrovasculature and microvasculature from damage [233,234,236], and ameliorates cardiac remodeling postmyocardial infarction [103]. The administration of FTY720 has been reported to prevent the formation of fibrosis by stimulating anti-inflammatory and anti-oxidant mechanisms in the myocardium [237]. It also reduces cardiomyocyte mortality in a mouse model of postmyocardial infarction, cardiac remodeling, and dysfunction [103], and ameliorates apoptosis, inflammation, and oxidative stress in an experimental model of myocardial stress [57]. Overall, these actions would confer cardioprotection, and thus, might also constitute part of a therapeutic prevention approach.

The long-term effects of sustained FTY720 exposure to cells promote the internalization and uncoupling of the receptor from its G protein, thereby leading to a reduction in the number of S1PR1s on the cell surface [235,238]. As this compound is present in much lower concentrations in plasma than S1P [235], it has been proposed that a switch of endogenous S1P signaling balance from S1PR1 to S1PR2 and S1PR3 would then be favored; i.e., causing the contraction of arterial smooth muscle cells and leading to a transient increase in blood pressure [239,240,241]. Considering that S1PR1 and S1PR3 are responsible for the beneficial effects of FTY720 on the cardiovascular system, it could be hypothesized that the mechanism likely involves S1PR3 signaling. Consistent with this view, this compound does not act via S1PR2 [235]. Moreover, S1PR3 has been directly involved in S1P-mediated heart protection against experimental I/R in vivo [84].

The distribution of FTY720 in plasma lipoproteins or other plasma components has not yet been reported. Indeed, it is not yet known whether this compound is mainly transported by HDL, as it occurs with S1P. In the context of diabetes mellitus, it is also unknown whether FTY720 loading might be effective at refunctionalizing diabetic HDL.

However, the appearance of a transient, but significant, bradycardia in phase II and III clinical trials of FTY720 [31] and lymphopenia has obscured its therapeutic use. Interestingly, FTY720 failed to induce bradycardia in S1PR3-deficient mice [242], suggesting a role for cardiac S1PR3 in mediating FTY720 signaling. However, S1PR1 agonists that do not act via S1PR3 induce bradycardia in humans [235]. Thus, studies are needed to fully assess safety events related to the suitability of long-term FTY720 treatment.

#### 8.2.2. SEW2871

SEW2871 ((5-[4-phenyl-5-(trifluoromethyl)-2-thienyl]-3-[3-(trifluoromethyl) phenyl]-1,2,4-oxadiazole) is a S1PR1-selective agonist that is structurally unrelated to S1P and is less active. SEW2871 phosphorylation is not required for binding to the receptor. SEW2871 induces S1PR1 internalization and recycling [243]. Importantly, SEW2871 does not activate S1PR3 and does not cause bradycardia but induces lymphopenia in wild-type mice [244]. This compound has been as effective as S1P in conferring resistance against cellular injury during hypoxia [71]. SEW2871 future therapeutic applications have also been hindered by the induction of adverse reperfusion arrhythmias in isolated rat heart preparations [245].

### 8.3. Other Potential Pharmacological Strategies to Raise S1P Levels: Lipid-Modifying-Based Therapies

#### 8.3.1. Statins

While statin treatments have been successful in lowering LDL cholesterol in patients, they also elicits modest elevations in HDL cholesterol [246,247], and the efficacy of these drugs in modulating S1P signaling has yet to be determined. The pharmacological activation of endothelial cells with statins, such as simvastatin and pitavastatin, upregulates *S1PR1* expression [248,249], increasing HDL-induced eNOS activity and conferring endothelial protection. In this regard, it could be hypothesized that the anti-atherogenic effect of statins would partly involve the activation of S1PR1 signaling. Furthermore, the potential contribution of statin-mediated increases in plasma HDL-bound S1P to improving HDL cardioprotective properties also needs further research.

#### 8.3.2. PPARγ

PPARγ is reportedly involved in S1P metabolism [250,251,252,253,254]. However, the effect of PPARγ inducers on S1P homeostasis is controversial. Indeed, PPARγ agonism increased hepatic *Apom* mRNA [251] and S1P synthesis by upregulating SphK in one study [250], but decreased the hepatic and plasma abundance of ApoM and S1P in diet-induced obese mice in another study [254]. Conversely, the use of a PPARγ antagonist produced the opposite effect (i.e., an increase in both S1P and ApoM) in exposed cultured hepatocytes (HepG2 cells). Intriguingly, S1P increases PPARγ activity [253].

PPARγ induction also elicits a shift of S1P and ApoM from HDL towards ApoB-containing lipoproteins [144]. Despite this, S1P signaling remains enhanced, potentially indicating the participation of HDL-independent mechanisms. Further research is needed to clarify the effect of PPARγ signaling on plasma S1P and ApoM.

## 9. Concluding Remarks

Cardiometabolic diseases have major economic and social impacts, with serious morbidities and increased mortality. The prevalence of HF is increased in diabetic patients. The early detection and identification of patients at high risk for HF onset may facilitate favorable interventions via lifestyle and/or pharmacological treatment. Thus, the identification of new biomarker(s) for the early diagnosis of diabetes-mellitus-related cardiomyopathy to improve HF prediction is needed.

Over the last few years, circulating sphingolipids have been gaining interest as predictive biomarkers for cardiac and metabolic disorders. Consistently, recent findings have shown that altered circulating sphingolipids were associated with defective insulin signaling [255,256,257,258,259]. The ability of sphingolipids to be trafficked from tissue to tissue using plasma lipoprotein carriers would also reveal a role for interorgan crosstalk. Actually, systemic alterations in sphingolipids, such as those found in diabetes mellitus, may also open new diagnostic venues to improve the prediction of diabetic complications, including cardiac dysfunction [260]. In this context, one possible new biomarker is plasma S1P. S1P is a bioactive molecule that is thought to be beneficial in the occurrence and development of myocardial ischemia in different clinical and experimental models.

The dysregulation of S1P signaling and of the metabolic machinery involved in the control of intracellular S1P levels may be a common characteristic of a number of cardiovascular diseases. Notably, S1P metabolism may appear unbalanced in cardiometabolic diseases such as obesity and diabetes mellitus, even in the absence of specific defects in proteins involved in intracellular sphingolipid signaling. Moreover, the physiopathological meaning of systemic alterations in total S1P and its distribution in different carriers is still poorly understood. Thus, active research is needed. The therapeutic effects of structural analogs of S1P, such as FTY720, on heart function shown in recent human intervention studies have provided promising results, though these results might have been obscured by the occurrence of some side effects. The enrichment of the main vehicles (HDL) of S1P might open novel, promising, S1P-raising therapeutic strategies in the treatment of cardiometabolic complications, including diabetes-mellitus-related cardiomyopathies like DCM.

## Figures and Tables

**Figure 1 ijms-20-06273-f001:**
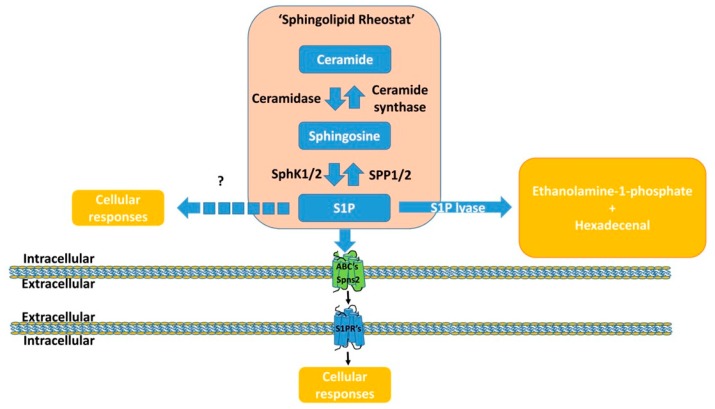
Illustration summarizing the cellular sphingolipid rheostat and signaling in mammalian cells. The intracellular balance among ceramide, sphingosine, and S1P species is defined as sphingolipid rheostat. Ceramide may be metabolized to sphingosine (Sph) by ceramidase. Sph can then be phosphorylated into S1P by SphK1/2. The produced S1P may directly influence cell physiology by yet unknown mechanisms. Intracellular content of S1P may be modulated by the actions of different enzymes. Indeed, it can be dephosphorylated by SPP1/2 or irreversibly degraded by S1P lyase to ethanolamine-1-phosphate and hexadecenal. S1P produced intracellularly may also be exported out of the cell via ATP-binding cassette (ABC) transporters (i.e., m ABCC1 and ABCG2) or its Spns2, depending on the cell type. Extracellular S1P can then regulate cellular functions by binding to one of its G-protein-coupled receptors (S1PR1–5). Abbreviations: ABC, ATP-binding cassette transporters; SM, sphingomyelin; Sph, sphingosine; SphK, sphingosine kinase; S1P, sphingosine-1-phosphate; SPP, sphingosine phosphate phosphatase; Spns2, spinster2.

**Figure 2 ijms-20-06273-f002:**
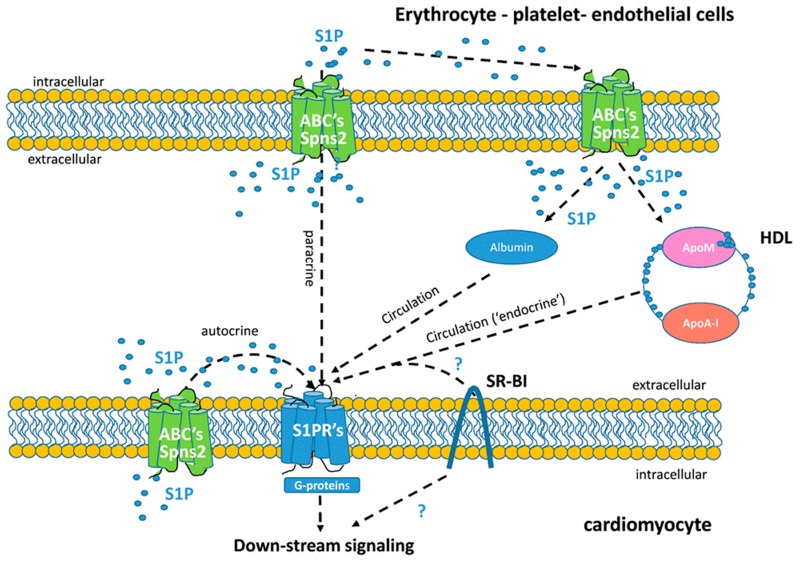
Schematic representation of mechanisms involved in inside-out trafficking of S1P. Schematic representation of autocrine, paracrine, and endocrine pathways involved in the cellular transport of S1P. S1P export may be mediated by different cell type-specific transporters, i.e., Spns2, and ATP-binding cassette (ABC) transporters, including ABCC1 and ABCG2. Extracellular S1P can then act on S1PR in autocrine, paracrine, and/or endocrine ways and promote cellular processes. HDL, and to a lesser extent, albumin, are regarded as the main carriers of S1P in plasma, and may mediate systemic S1P transport to target tissues. Abbreviations: ABC, ATP-binding cassette transporters; Apo, apolipoprotein; S1P, sphingosine-1-phosphate; S1PRs, S1P receptors; SR-BI, scavenger receptor class B type 1; Spns2, Spinster2.

**Figure 3 ijms-20-06273-f003:**
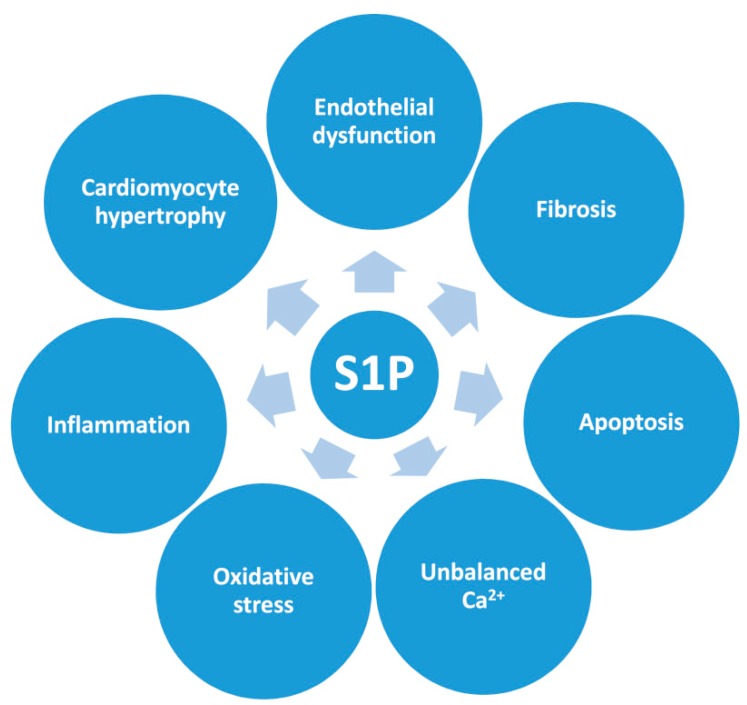
Hypothetical diagram showing how HDL-bound S1P may favorably influence different cell processes frequently altered in diseased hearts. S1P—sphingosine-1-phosphate.

**Figure 4 ijms-20-06273-f004:**
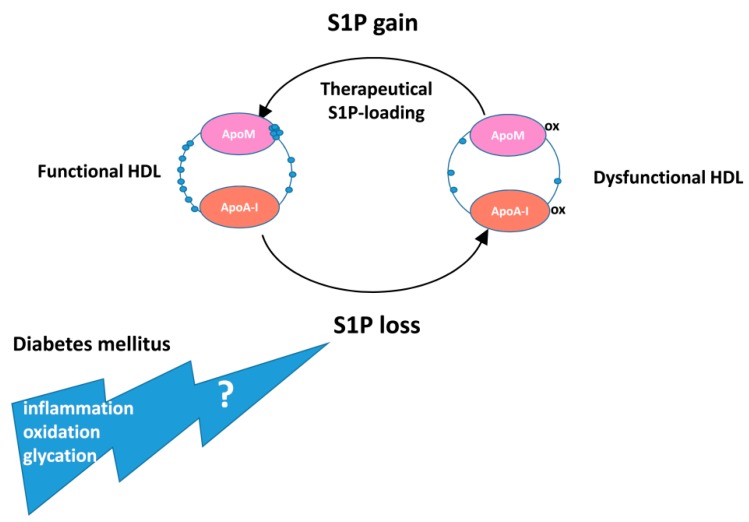
Therapeutic potential of S1P loading of HDL in diabetes mellitus. Schematic model showing how S1P loss may negatively influence HDL function in diabetes mellitus, and the therapeutic potential provided by their replenishment with S1P aimed at refunctionalizing diseased HDL.

**Figure 5 ijms-20-06273-f005:**
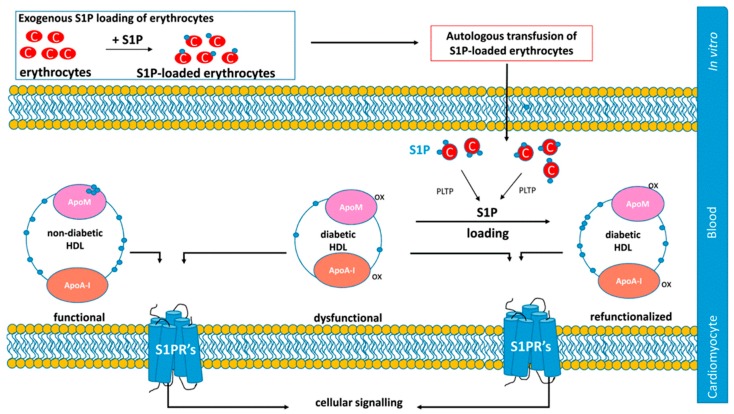
Therapeutic use of transfusions of autologous S1P-preloaded erythrocytes to elevate HDL-bound S1P for treating cardiac diseases. The proposed strategy was first described by Sattler et al., (2015) [56]. Briefly, isolated erythrocytes may be freshly isolated and loaded ex vivo with S1P. S1P-reloaded erythrocytes may be autologously transfused to the same donor. Circulating S1P-loaded erythrocytes may transfer S1P with the circulating HDL, possibly involving the action of ABC transporters and/or Spns2 and PLTP. S1P-reloaded HDL may then contribute to restoring HDL dysfunction. This figure has been adapted from [56].

**Figure 6 ijms-20-06273-f006:**
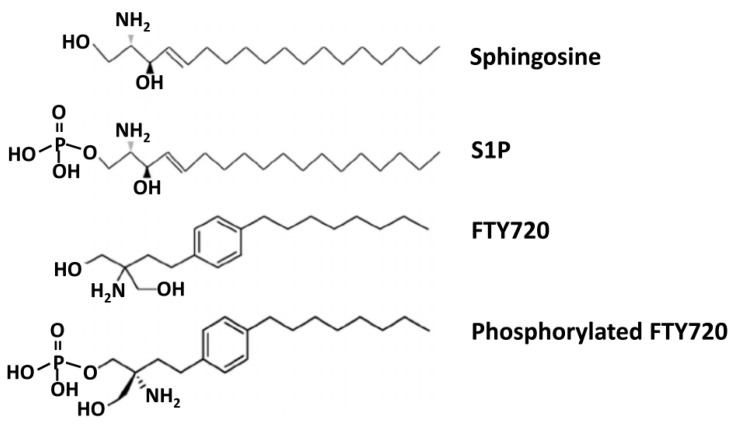
Structural analogies between FTY720 and S1P. Molecular structures of sphingosine, S1P, FTY720, and phosphorylated FTY720 are shown.

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
