# Peer review of "Novel Insights into the Role of HDL-Associated Sphingosine-1-Phosphate in Cardiometabolic Diseases"

_ijms, 2019, doi:10.3390/ijms20246273_

Round 1

Reviewer 1 Report

The review work provided by Diarte-Añzaco et al encompasses a wide array of recent progresses made in the elucidation of the role of HDL-associated sphingosine-1-phosphate in cardiometabolic disease, offering a valuable material for novice to read and obtain a quick master of studies in this filed. This review is well organized and written. However, the tenses used in the text are not coherent. In addition, the authors seems inadvertently miss some literatures, please add them as suggested. Overall, the review is almost ready to prime.

Specific points:

Please check the labels of SPP1/2 and SphK1/2. Based on the description in the figure legends, the positions of those two labels should be swapped.

Line 163. Please explain the “coral action”. Line 185. add a reference Line 191. are commonly involved in… Line 192. More important works should be cited in line 192 or in the figure legend 3 to support the involvement of the so-called cellular mechanisms. Line 201-205.Salient works should be cited. Line 207. supports Line 228. change “was” to “is”

         In addition, references are lacking.

Line 229. Please change “promoted” to “promote” for coherence. Line 233-234 Narrative is quite confusing. Please rephrase it. Line 256. double-check [i] need references for in vivo studies Line 274. delete “of” Line 277. “favorably regulates” is of no problem in grammar. However, this description is less informative. Please use more specific terms. Line 311. exert influence on Line 312. abrogate Line 320 form Line 311. change “is” to “are” Line 324 please specify what kinds of effects are induced by SIP. Line 337. Please add pertinent references Line 341. suggest Line 364-36. Need references Line 400. Reference is lacking Line 408. does not Line 414. Whether it is mediated by… Line 421. suggest Line 436-440. Double check the tenses used in those lines Line 451. is Line 460. suggest Line 492. prevail Line 500. fails Line 487-508. Double check the tenses. Ditto for line 514-524 Line 530. is Line 531. is Line 547. lead Line 602 based on Line 614. increases Line 621. Writing is very confusing. Line 640-641. Check the writing Line 652. has Line 654. is

Author Response

Reviewer 1

The review work provided by Diarte-Añzaco et al encompasses a wide array of recent progresses made in the elucidation of the role of HDL-associated sphingosine-1-phosphate in cardiometabolic disease, offering a valuable material for novice to read and obtain a quick master of studies in this filed. This review is well organized and written. However, the tenses used in the text are not coherent. In addition, the authors seems inadvertently miss some literatures, please add them as suggested. Overall, the review is almost ready to prime.

We fully appreciate this reviewer for the time taken and provided comments and recommendations to improve this review. As described in detail below, the indications related to English style/grammar and lacking references have been addressed. English style has been revised by American Journal Experts. Suggested changes to improve understanding as well as the mistake detected in Figure 1 have also been addressed.

We the authors sincerely hope that the revised ms. will now meet reviewer’s requirements.

Specific  points:

Please check the labels of SPP1/2 and SphK1/2. Based on the description in the figure legends, the positions of those two labels should be swapped.

Figure 1 has been updated and the labels SPP1/2 and SphK1/2 interchanged.

Line 163. Please explain the “coral action”.

The word ‘coral’ was mistyped; the correct word was actually choral, which would mean ‘the orchestrated action of many factors at the same time’. It has been substituted by the word ‘combined’ to avoid any misunderstanding.

As requested, orthographic/grammatical/English style/confusing narrative indicated in lines 191, 207, 229, 233-234, 274, 277, 311, 312, 320, 321, 341, 408, 414, 421, 436-440, 460, 492, 500, 487-508, 514-524, 530, 531, 547, 602, 614, 621, 640-641, 652, 654 has been properly addressed in the revised version of the ms.

As requested, suitable  more important references have been mentioned in lines 185, 192, 201-205, 337, 3643-365, 400 in the revised version of the ms.

Line 256. double-check [i] need references for in vivo studies

Reference 85 mentions the in vivo study; reference caption has been properly located to avoid confusion.

Line 324 please specify what kinds of effects are induced by SIP.

As requested, the specific effects elicited by S1P have been mentioned.

Reviewer 2 Report

Authors reviewed the implication of sphingolipids, specifically sphingosine-1-phosphate, in the cardiomyopathy of cardiometabolic diseases such as diabetes mellitus (diabetic cardiomyopathy). The authors take in consideration the potential role of these molecules as circulating biomarkers in the diagnosis of main cardiometabolic complications frequently associated with systemic metabolic syndromes, and evaluate its therapeutic potential for the treatment of cardiometabolic diseases.

It is a quite exhaustive revision of the bibliography and only some aspects are a bit confusing.

In lane 97, the sentence “two types of sphingomyelinases may hydrolyze sphingomyelin yielding ceramide” oposses the sentence in lane 105 “Ceramide can be transformed reversibly into SM by sphingomyelinase”. Please, clarify which one is correct and provide a reference for the statement in the text, because reference 22 describes SPP1, not sphingomyelinases. Label ER in Figure 1. In this figure, SPP1/2 and SphK1/2 are in the opposite way. Please, define the abbreviation ABCs in the caption. Figure 2 is quite complex. It seems that Panel A is redundant with Panel B. In Panel B, what does it mean SR-BI? It is not describe in the caption but even not in text, only much more below. The direct pass of S1P through plasma membranes is not explained in text. As it is stated, for example in reference 29, S1P cannot freely pass through plasma membranes. Therefore, if that is a real possibility, please, refer to in text even with bibliography. In lane 169, references should be removed from there and pass them to lane 168 in between references 50-60. Should small circles in Figure 3 contain the same processes as those explained below? Also, font inside them should be larger. Subtitles from lane 200 and so on maybe should be numbered, following the shape of section 3, as 4.1, 4.2, … The same in lane 428, section 5.3. In lane 208, reference 73 is an editorial commenting on the original article that should be the correct reference (75). In lanes 211, 213, should say “inhibition” not “degradation”. In lane 307, reference 124 is about astrocytes, not cardiac cells. In lane 366, what does it mean “has recently been reported S1P efluxed to plasma HDL”? Was not that said in 2000 and 2007? (references 28, 147 and 148). In lane 406, SR-B1 is not defined but maybe it can be removed because in the following sentence is only regarding ABCA1 receptor. Reference 178 seems to have nothing to do with the manuscript In lane 411, define SR-BI. Lane 525. The sentence “Whether plasma levels of S1P may be useful in predicting increased risk for cardiovascular disease in diabetic patients is still unknown” should be modified because in lane 532-533 is make it clear that “plasma S1P has also been identified as an independent predictor against the development of coronary artery disease in patients with type diabetes mellitus [208]”. In addition, the next two sentences are quite confused. In Figure 5, what is the meaning of the upper membrane? What is separating? Is it not the direct pass of erythrocytes to the blood flow by a transfusion? Lane 621. Why should be interesting to block S1P effects, by blocking S1PRs? This contrasts with the strategies to raise S1P levels to treat cardiac diseases (see above in the manuscript). Please, clarify. Remove apostrophe in plurals, like in SphKs, SPPs, ABCs, S1PRs, …

Minor comments:

Because manuscript has been corrected by American Journal Experts, several existing errors should be typographical errors but must be corrected:

Lane 34, “the therapeutical potential”. Lane 52, “altered balances”. Lane 60, “the relationship”. Lane 111, “or Spns2”. Lane 137, “that may regulate”. Lane 162, “Which of these S1PRs” or “Whether one of these S1PRs”. Lanes 172-173, reconsider remove one “involved” in the sentence. Lane 185, “development of DCM” or “DCM development”. Lane 189, “S1P elevations” or “elevations of/on S1P”. Lane 191-192, “The cellular mechanisms … are thought to mainly”. Lanes 197-198, remove one “different”. Lane 207, “supports”. Lane 214, “decrease has” or “decreases have”. Lane 222, “treatment with S1P”? Lane 233, “reduced the extent”. Lane 274, remove “of”. Lane 293, “the inhibition of an endogenous SphK1 inhibitor, …, conferred concomitant protection cardiomyocytes from apoptosis” maybe should be “… conferred concomitant protection from apoptosis to cardiomyocytes”? or “conferred cardiomyocytes concomitant protection from apoptosis”? Lanes 311-312, maybe is “favorable influence in myocardium fibrosis”? Lane 320, “form”. Lane 321, “cells are”. Lane 344, “suggests”. Lane 379, “do not”. Lanes 385-386, do you mean “In these ApoM transgenic mice, S1P synthesis was enhanced, thereby …”? Lane 390, maybe “catalyzes” is not the right term because Spns2 is a transporter and does not perform an enzymatic reaction. Lane 391, “influences”. Lane 415, “coordinating”. Lane 418, “Supporting this,”? Lane 420, “S1P-mediated”. Lane 520, “is decreased in a study with type 2 diabetes mellitus”. Lane 533, add a period after [208]. Lane 564, “respect to controls”. Lane 580, maybe remove “needs”? Lane 594, “transporters”. Lane 609, “mechanism of action, efficacy”. Lane 641, “oxadiazole) is a S1PR1-selective agonist”. Lane 650, remove parenthesis. Lane 663, “HepG2 cells”. Lane 688, “including diabetes mellitus-related cardiomyopathies like DCM” or maybe is quite redundant.

Author Response

Reviewer 2

Authors reviewed the implication of sphingolipids, specifically sphingosine-1-phosphate, in the cardiomyopathy of cardiometabolic diseases such as diabetes mellitus (diabetic cardiomyopathy). The authors take in consideration the potential role of these molecules as circulating biomarkers in the diagnosis of main cardiometabolic complications frequently associated with systemic metabolic syndromes, and evaluate its therapeutic potential for the treatment of cardiometabolic diseases.

It is a quite exhaustive revision of the bibliography and only some aspects are a bit confusing.

We sincerely thank the reviewer for the comments and recommendations made to improve this work. As described in detail below, indications related to English style/grammar, lacking references or uncompleted captions have properly been addressed in the revised version of the ms. Suggested changes to improve understanding as well as the mistake detected in figure 1 have been also addressed.

In lane 97, the sentence “two types of sphingomyelinases may hydrolyze sphingomyelin yielding ceramide” oposses the sentence in lane 105 “Ceramide can be transformed reversibly into SM by sphingomyelinase”. Please, clarify which one is correct and provide a reference for the statement in the text, because reference 22 describes SPP1, not sphingomyelinases. Label ER in Figure 1. In this figure, SPP1/2 and SphK1/2 are in the opposite way. Please, define the abbreviation ABCs in the caption.

As suggested, caption in figure 1 legend has been changed to precisely describe the main focus of this figure, showing the ‘sphingolipid rheostat’. As required, reference 22 has been substituted. Moreover, Figure 1 has also been corrected as SPP1/2 and SphK1/2 labels were mistakingly swapped in the previous version of this ms. Accordingly, ABC spelling has been added in this same figure caption.

Figure 2 is quite complex. It seems that Panel A is redundant with Panel B. In Panel B, what does it mean SR-BI? It is not describe in the caption but even not in text, only much more below. The direct pass of S1P through plasma membranes is not explained in text. As it is stated, for example in reference 29, S1P cannot freely pass through plasma membranes. Therefore, if that is a real possibility, please, refer to in text even with bibliography.

Panel A of figure 2 has been removed from the figure. The scheme in panel B, current figure 2, has been corrected. In this regard, the transport of S1P though plasma membranes has been removed from the figure.

In lane 169, references should be removed from there and pass them to lane 168 in between references 50-60.

Done.

Should small circles in Figure 3 contain the same processes as those explained below? Also, font inside them should be larger.

Subtitles from lane 200 and so on maybe should be numbered, following the shape of section 3, as 4.1, 4.2, … The same in lane 428, section 5.3.

As requested, subsections of section 4 have been numbered.

In lane 208, reference 73 is an editorial commenting on the original article that should be the correct reference (75).

Done.

In lanes 211, 213, should say “inhibition” not “degradation”.

Done.

In lane 307, reference 124 is about astrocytes, not cardiac cells.

The sentence and reference has been removed.

In lane 366, what does it mean “has recently been reported S1P efluxed to plasma HDL”? Was not that said in 2000 and 2007? (references 28, 147 and 148).

Done.

In lane 406, SR-B1 is not defined but maybe it can be removed because in the following sentence is only regarding ABCA1 receptor.

SR-BI mention has been removed in the revised version of the ms.

Reference 178 seems to have nothing to do with the manuscript

Reference 178 has been removed from the ms.

In lane 411, define SR-BI.

SR-BI abbreviation has been indicated in the revised version of the ms.

Lane 525. The sentence “Whether plasma levels of S1P may be useful in predicting increased risk for cardiovascular disease in diabetic patients is still unknown” should be modified because in lane 532-533 is make it clear that “plasma S1P has also been identified as an independent predictor against the development of coronary artery disease in patients with type diabetes mellitus [208]”. In addition, the next two sentences are quite confused.

This paragraph has been corrected.

In Figure 5, what is the meaning of the upper membrane? What is separating? Is it not the direct pass of erythrocytes to the blood flow by a transfusion?

In this figure, a schematic representation of a vessel is intended; membranes would belong to the luminal surface of endothelial cells.

Lane 621. Why should be interesting to block S1P effects, by blocking S1PRs? This contrasts with the strategies to raise S1P levels to treat cardiac diseases (see above in the manuscript).

This controversy has been clarified in the corresponding section. It is believed that S1P or FTY720 would elicit its favorable response via other receptors.

Since discovery of FTY720, new-generation S1P receptor drugs are being investigated to target S1P receptors, and to avoid inhibition of S1PR signalings. Overall, the family of S1PRs thus appears worthy of continued study and may provide significant therapeutic opportunities.

Please, clarify. Remove apostrophe in plurals, like in SphKs, SPPs, ABCs, S1PRs, …

Done.

Minor comments:

Because manuscript has been corrected by American Journal Experts, several existing errors should be typographical errors but must be corrected:

Lane 34, “the therapeutical potential”. Lane 52, “altered balances”. Lane 60, “the relationship”. Lane 111, “or Spns2”. Lane 137, “that may regulate”. Lane 162, “Which of these S1PRs” or “Whether one of these S1PRs”. Lanes 172-173, reconsider remove one “involved” in the sentence. Lane 185, “development of DCM” or “DCM development”. Lane 189, “S1P elevations” or “elevations of/on S1P”. Lane 191-192, “The cellular mechanisms … are thought to mainly”. Lanes 197-198, remove one “different”. Lane 207, “supports”. Lane 214, “decrease has” or “decreases have”. Lane 222, “treatment with S1P”? Lane 233, “reduced the extent”. Lane 274, remove “of”. Lane 293, “the inhibition of an endogenous SphK1 inhibitor, …, conferred concomitant protection cardiomyocytes from apoptosis” maybe should be “… conferred concomitant protection from apoptosis to cardiomyocytes”? or “conferred cardiomyocytes concomitant protection from apoptosis”? Lanes 311-312, maybe is “favorable influence in myocardium fibrosis”? Lane 320, “form”. Lane 321, “cells are”. Lane 344, “suggests”. Lane 379, “do not”. Lanes 385-386, do you mean “In these ApoM transgenic mice, S1P synthesis was enhanced, thereby …”? Lane 390, maybe “catalyzes” is not the right term because Spns2 is a transporter and does not perform an enzymatic reaction. Lane 391, “influences”. Lane 415, “coordinating”. Lane 418, “Supporting this,”? Lane 420, “S1P-mediated”. Lane 520, “is decreased in a study with type 2 diabetes mellitus”. Lane 533, add a period after [208]. Lane 564, “respect to controls”. Lane 580, maybe remove “needs”? Lane 594, “transporters”. Lane 609, “mechanism of action, efficacy”. Lane 641, “oxadiazole) is a S1PR1-selective agonist”. Lane 650, remove parenthesis. Lane 663, “HepG2 cells”. Lane 688, “including diabetes mellitus-related cardiomyopathies like DCM” or maybe is quite redundant.

As suggested, all the above grammar/typographical errors have been corrected. English style has been revised by American Journal Experts.

Reviewer 3 Report

The authors summarize the current literature on the potential impact of HDL-bound S1P-raising strategies for the treatment of cardiometabolic diseases.

The review is well-written and clear. The authors provide a great summary of the literature related to the topic. However, the review reads more like a report, rather than a review, as there is limited discussion of gaps in the field and hypotheses of what research needs to be done to advance the field. I only suggest two point:

The authors should provide personal discussion and interpretation of summarized information at the end of main paragraphs. Please, briefly introduce S1P in paragraph 1.

Author Response

Reviewer 3

The authors summarize the current literature on the potential impact of HDL-bound S1P-raising strategies for the treatment of cardiometabolic diseases.

The review is well-written and clear. The authors provide a great summary of the literature related to the topic. However, the review reads more like a report, rather than a review, as there is limited discussion of gaps in the field and hypotheses of what research needs to be done to advance the field. I only suggest two point:

The authors should provide personal discussion and interpretation of summarized information at the end of main paragraphs. Please, briefly introduce S1P in paragraph 1.

As requested, we have deepened the discussion on unknown issues that in our opinion need further research. Added text has been highlighted in yellow. In this regard, sections 8 and 9 have been updated to the best of our knowledge.

It should be considered that there is a lack of appropriate models to investigate the contribution of a dysregulated sphingolipid metabolism in the context of cardiac diseases with a common origin, such that provided by diabetes mellitus. Although most of compelling evidence supports the notion that an altered sphingolipid pattern, directly contributes to the development of cardiovascular disease, the number of reports checking the potential S1P as a biomarker is still limited.